# Adoptive Cell Immunotherapy in Relapse/Refractory Epstein–Barr Virus-Driven Post-Transplant Lymphoproliferative Disorders

**DOI:** 10.3390/antib14020047

**Published:** 2025-06-12

**Authors:** Martina Canichella, Paolo de Fabritiis

**Affiliations:** 1Hematology, St. Eugenio Hospital, ASL Roma2, 00144 Rome, Italy; paolo.defabritiis@aslroma2.it; 2Department of Biomedicina e Prevenzione, Tor Vergata University, 00133 Rome, Italy

**Keywords:** PTLD—post-transplant lymphoproliferative disorders, RIS—reduction in immunosuppression, EBV—Epstein–Barr virus, allo-HSCT—allogeneic stem cell transplantation

## Abstract

Post-transplant lymphoproliferative disorders (PTLD) represent a life-threatening complication following solid organ transplantation (SOT) and allogeneic hematopoietic stem cell transplantation (allo-HSCT), particularly in patients with relapsed or refractory (R/R) disease, where therapeutic options are limited and prognosis is poor. Among emerging strategies, adoptive cellular immunotherapy—specifically Epstein–Barr virus-specific cytotoxic T lymphocytes (EBV-CTLs)—significantly improved outcomes in this challenging patient population. EBV-CTLs restore virus-specific immunity and induce sustained remissions with minimal toxicity, even in heavily pretreated individuals. The most promising cellular product to date is tabelecleucel, an off-the-shelf, allogeneic EBV-specific T-cell therapy, which is currently the only cellular therapy approved by the European Medicines Agency (EMA) for the treatment of R/R EBV-positive PTLD following SOT or allo-HSCT. This review aims to provide an overview of PTLD treatment with a specific focus on adoptive cellular immunotherapy. We highlight the most robust clinical outcomes reported with EBV-CTLs, particularly those achieved with tabelecleucel, and explore emerging cellular approaches such as CAR T-cell therapy, which may further broaden therapeutic strategies in the near future.

## 1. Introduction

Post-transplant lymphoproliferative disorders (PTLD) are rare but serious complications that can develop after transplantation—including allogeneic stem cell transplantation (allo-HSCT) and solid organ transplantation (SOT)—and they are strongly related to immunosuppression conditions [1]. PTLD includes several groups of B-cell proliferative disorders, classified by the World Health Organization (WHO) into six categories: three non-destructive types (plasmacytic hyperplasia, infectious mononucleosis-like PTLD, and florid follicular hyperplasia), as well as polymorphic PTLD, monomorphic PTLD, and a form resembling classic Hodgkin lymphoma [2]. Most cases of PTLD (52% to 80%) are driven by Epstein–Barr virus (EBV) reactivation in latent B cells. EBV is involved in nearly all cases of PTLD following allo-HSCT and in half of those after SOT. Several risk factors for EBV-related PTLD have been identified: in SOT recipients, risk factors include the type and intensity of immunosuppression, the transplanted organ, and mismatched EBV serostatus between donor and recipient. In particular, the risk of developing PTLD post-SOT varies by the type of organ transplanted, with the highest risk observed in intestinal transplants, followed by lung, heart, liver, pancreas, and kidney [3]. In allo-HSCT, key risk factors include EBV seromismatch, T-cell depletion, use of haploidentical or cord blood donors, reduced-intensity conditioning (RIC), older recipient age, anti-thymocyte globulin (ATG) or anti-CD3 therapy, and prolonged immunosuppression, especially in the context of graft-versus-host disease (GvHD) management.

First-line treatment for PTLD post-SOT involves the reduction in immunosuppression (RIS) followed by systemic therapy (rituximab, anti-CD20 monoclonal antibody, and ±chemotherapy), while in post-allo-HSCT cases, rituximab—with or without RIS—is the preferred front-line approach. As a second-line option, the NCCN Clinical Practice Guidelines recommend EBV-specific cytotoxic T-lymphocyte (EBV-CTL) therapy [4]. Other drugs and therapeutic strategies are currently under investigation. Among these, the use of CAR-T cell therapy appears to be a promising option, although its application is still limited by several challenges and limitations, as discussed below. In this review we briefly illustrated the first-line treatment of PTLD and focused on the emerging cell-based treatment in R/R PTLD. In particular, we reported the experience with tabeleucleucel and chimeric antigen receptor T cell (CAR-T).

## 2. First-Line Treatment

The treatment of PTLD can be summarized into three main strategies: the reduction in B-cell burden, the restoration of T-cell immunocompetence, and the antiviral treatment (Figure 1). It is important to point out that the efficacy of antiviral treatment in PTLD is still limited. Antiviral nucleoside analogs, such as ganciclovir and acyclovir, require the lytic phase to be activated. In PTLD, the latency of EBV could induce resistance to these antiviral agents, leading to unsatisfactory anti-lymphoma activity.

### 2.1. Frontline Treatment Post-SOT

In post-SOT PTLD, RIS is the initial and essential therapeutic step [6,7]. There are no universally accepted guidelines for RIS, and approaches should be tailored based on disease burden and graft type. Some centers adopt the strategy used in the phase II PTLD-1 trial, in which mycophenolate mofetil and azathioprine are discontinued, calcineurin inhibitor doses are reduced by 30–50%, and corticosteroid doses are either maintained or reduced. Two to four weeks after initiating RIS, imaging is performed to assess treatment response. If complete remission (CR) is achieved, no additional therapy is required. However, in cases of stable disease (SD) or progression, rituximab monotherapy is commonly introduced as the next line of treatment [6,7].

In addition to RIS, systemic therapy with rituximab is indicated for most patients with monomorphic CD20-positive PTLD (mPTLD). Rituximab is administered once weekly for up to four doses and demonstrated to improve progression-free survival (PFS) and overall survival (OS). The first prospective study on the use of rituximab was reported by Choquet et al. In this study, 43 patients who failed to achieve CR following RIS were subsequently treated with weekly rituximab. An overall response rate (ORR) of 44% was observed, with a CR rate of 27%. The 1-year OS was 67%, and the median OS was 15 months [8].

Subsequently, a Spanish study demonstrated that patients achieving only PR benefited from the addition of 2 to 4 further cycles of rituximab, suggesting a potential role for extended therapy in this subgroup [9].

The basis of the sequential and risk stratification strategy of PTLD post-SOT has been established by three main phase 2 clinical trials: PTLD-1 (NCT01458548), PTLD-1/3 (NCT00590447), and PTLD-2 (NCT02042391). The PTLD-1 trial provided robust evidence supporting the efficacy and tolerability of a sequential approach consisting of four cycles of intravenous (IV) rituximab monotherapy, followed by four cycles of CHOP (cyclophosphamide, doxorubicin, vincristine, and prednisone every 21 days) chemotherapy in patients who failed to achieve CR after RIS [10]. Notably, this stepwise treatment strategy was associated with a more favorable safety profile and resulted in a markedly improved median OS of 6.5 years, substantially exceeding the 2.4-year median OS previously observed with rituximab monotherapy alone. Importantly, patients who exhibited a clinical response to rituximab experienced the most favorable outcomes. The PTLD 1/3 trial further refined treatment stratification after four infusions of intravenous rituximab, distinguishing patients in CR—who were subsequently managed with rituximab consolidation—from those who failed to achieve CR and therefore required escalation to four cycles of R-CHOP [11]. This approach not only supported a response-adapted strategy but also highlighted a subset of patients with particularly poor prognoses. In addition to those with primary resistance to rituximab, further sub-analysis identified high-risk features, including an International Prognostic Index (IPI) score greater than 2 and recipients of thoracic solid organ transplants (SOT), as predictors of inferior outcomes. The PTLD-2 trial evaluated response and risk-adapted treatment approach, escalating therapy in high-risk patients and de-escalating in those at low risk [12]. Risk was stratified based on response to initial rituximab, IPI score, and type of organ transplantation. Low-risk patients (CR or PR with IPI ≤ 2) received rituximab consolidation alone, achieving a 2-year OS of 100%. High-risk patients (PR with IPI > 2, SD, or progression without thoracic SOT) received R-CHOP, with outcomes comparable to PTLD-1. Very high-risk patients (thoracic SOT with progression) received intensified chemoimmunotherapy (R-CHOP/dexamethasone, high-dose cytarabine, and oxaliplatin (DHAOX)), with limited efficacy (2-year OS 30%). These findings suggest that a subset of patients with PR and favorable risk features may avoid chemotherapy, while escalation beyond R-CHOP offers little benefit in high-risk patients.

### 2.2. Frontline Treatment of PTLD Post-Allo-HSCT

In allo-HSCT-associated PTLD, RIS is frequently limited by the risk of GvHD. Consequently, first-line treatment for allo-HSCT-PTLD typically involves rituximab, with or without RIS when feasible. Response rates to frontline rituximab monotherapy remain suboptimal, with CR observed in approximately 20% of cases, ORR ranging from 60% to 65%, and a 2-year OS of around 50% [13,14]. Over the years, for R/R PTLD, EBV-CTLs emerged as an effective and safe strategy (Figure 1).

## 3. Cell-Based Therapy

R/R PTLD following SOT or allo-HSCT remains an area of significant unmet clinical need, with poor OS outcome. Median survival has been estimated at approximately 0.7 months post-allo-HSCT and 4.1 months post-SOT [15,16,17]. No standardized treatment approach is currently established for these patients. In EBV-driven cases, the use of EBV-specific cytotoxic T lymphocytes (EBV-CTLs) is recommended [18].

## 4. EBV-Specific Cytotoxic T Lymphocytes

The adoptive immunotherapy based on EBV-specific T lymphocytes has always represented an attractive option for the treatment of PTLD (Table 1). The rationale of this approach is the adoptive transfer of EBV-specific lymphocytes to transplant recipients, aiming to target and eliminate EBV-infected B cells. This therapeutic strategy encompasses both donor lymphocyte infusions (DLIs) and EBV-specific CTLs, which may be derived from either the original transplant donor or a third-party EBV-seropositive individual [18]. The administration of DLI has historical value due to the high risk of GvHD.

The first clinical evidence supporting the use of EBV-CTLs in transplant recipients dates back to 2002, when Comoli et al. described seven individuals (comprising four pediatric and three adult patients) with high EBV-DNA titers (defined as 1000 genome copies of EBV-DNA per 10^5^ cells in two consecutive samples) who received autologous EBV-specific CTLs. The infusions were well tolerated, and no episodes of organ rejection were documented. These observations provided a rationale for exploring autologous CTLs in both prophylactic and preemptive contexts [24]. In the same year, Haque et al. [25] reported the result of the treatment with partly HLA-matched allogeneic EBV-CTLs from healthy donors to treat R/R EBV-driven PTLD. Eight SOT recipients received infusions that demonstrated both therapeutic efficacy and minimal toxicity [25]. In 2007, a multicenter phase II clinical trial assessed the efficacy of allogeneic EBV-CTLs in 33 patients (31 SOT and 2 allo-HSCT recipients) with relapsed or refractory PTLD. These CTLs were derived from EBV-seropositive donors. The ORR was 64% at 5 weeks and 52% at 6 months. Importantly, no events of GvHD, organ rejection, cytokine release syndrome (CRS), or neurotoxicity were observed [26]. These encouraging outcomes, along with improved manufacturing protocols and HLA matching strategies, paved the way for the development of tabelecleucel. Tabelecleucel is an off-the-shelf, allogeneic product composed of polyclonal EBV-specific T cells sourced from healthy, EBV-seropositive donors and selected on HLA compatibility with the recipient. Over the last two decades, numerous studies have confirmed its favorable safety profile and clinical activity. Notably, in 2020, Prockop et al. published data on 46 patients—33 SOT and 13 allo-HSCT recipients—with EBV-associated PTLD unresponsive to rituximab. Third-party, HLA-restricted EBV-CTLs were administered as three weekly infusions followed by a three-week monitoring period [27]. Complete or sustained partial responses were observed in 68% of allo-HSCT recipients and 54% of SOT recipients. Among those achieving CR/PR or stable disease after the initial treatment cycle, one-year OS resulted in 88.9% and 81.8%, respectively.

The same authors later reported the aggregate OS across the combined population of patients with EBV + PTLD following allo-HSCT and SOT, considering the cases with the best overall response (BOR = CR or PR) treated with tabelecleucel [28].

The authors considered the OS achieved in three clinical trials (NCT00002663, NCT01498484, and NCT02822495). 48/76 (63.2%) patients achieved BOR, including 32 pts with CR and 16 pts with PR. After a median follow-up of 14.8 months (0.4–115.0 months), the median OS resulted in 54.6 months. Considering the entire study population, the estimated 1- and 2-year OS rates were 65.8% and 57.8%, respectively. According to the responders (CR + PR, *n* = 48), the estimated 1- and 2-year OS rates were 91.3% and 86.2%, respectively. Two-year estimated OS rates were 86.2% and 86.5% for patients with CR and PR, respectively [28]. These results showed that the 2-year OS of patients who achieved PR was similar to the 2-year OS of those who had CR. Overall, the treatment was well tolerated with no evidence of GvHD, rejection, CRS, or neurological symptoms. Based on this evidence, in 2022, tabelecleucel received EU approval as the first allogeneic T-cell immunotherapy (Ebvallo^®^) [29].

The primary body of evidence supporting the efficacy and safety of tabelecleucel is derived from the ALLELE study (NCT03394365), a multicenter, single-arm, phase III study that enrolled patients with R/R EBV + PTLD following allo-HSCT or SOT [30]. Between 27 June 2018 and 5 November 2021, a total of 63 patients were screened, 43 of whom received at least one dose of tabelecleucel—14 in the allo-HSCT cohort and 29 in the SOT group. The treatment regimen consisted of tabelecleucel administered at 2 × 10⁶ cells/kg on Days 1, 8, and 15 of each 35-day cycle. An objective response was achieved in 7 of 14 patients (50%) in the allo-HSCT arm and in 15 of 29 (52%) in the SOT arm, with a median follow-up of 14.1 and 6 months, respectively. Notably, this trial was the first to report the median duration of response (DOR) to tabelecleucel, which reached 23 months in the allo-HSCT group and 15.2 months among SOT recipients. The DOR across the entire study population was 23 months. Importantly, no cases of GvHD, organ rejection, or fatal treatment-emergent serious adverse events (TESAEs) were observed. Additionally, no instances of tumor flare were reported; this was an encouraging finding in contrast to the previously noted safety signals from a broader pooled analysis involving 180 patients with R/R EBV+ PTLD treated with tabelecleucel.

An updated analysis of the ALLELE trial’s efficacy and safety data was presented at the 2025 EBMT Annual Meeting in Florence [31]. This latest dataset included 75 patients, 26 post-allo-HSCT and 49 post-SOT. Clinical characteristics (median age, ECOG), PTLD histological subtype, and number and type of prior lines of therapy were similar between the post-SCT and post-SOT subgroups. The ORR (CR + PR) was 50.7%, with 21 patients (28.0%) achieving a complete response and 17 (22.7%) a partial response. Among responders, the 12-month PFS rate was 71.9%. Moreover, the 1-year OS rate was markedly higher in responders compared to non-responders (78.2% vs. 55.7%). The median OS for the overall cohort was 18.4 months. Once again, tabelecleucel maintained a favorable safety profile, with no reports of fatal TESAEs, GvHD, or transplant rejection. Another area of application of tabelecleucel is the setting of PTLD with central nervous system (CNS) involvement, a form with very few treatment options. A study regarding a cohort of 18 patients with PTLD and CNS involvement reported an ORR of 77.8% (14 out of 18), with CR and PR of 38.9% achieved in both cases (*n* = 7) [32]. The median time to response was 1.8 months, ranging from 0.7 to 6.4 months. At one and two years, OS rates were 70.6% and 54.9%, respectively. Notably, survival outcomes were markedly better in responders, with 1-year and 2-year OS reaching 85.7% and 66.7%, compared to 0% for non-responders at both time points. A phase 2 clinical trial (EBVision-ATA129-EBV-205) is currently underway to further assess the therapeutic potential of tabelecleucel in patients with EBV-positive malignancies, including those with CNS-involved EBV PTLD.

## 5. CAR-T

In the last two decades, anti-CD19 chimeric antigen receptor (CAR) T-cell therapy has emerged as a game changer, particularly in the treatment of R/R non-Hodgkin B-cell lymphoma [5,33,34]. In these settings, CAR-T cells redefined the therapeutic algorithm and subsequently improved the prognosis.

Since PTLD is most often a B-cell lymphoma, CAR-T cells have been applied in resistant cases, yielding promising results [35]. However, the use of CAR-T cell therapy in PTLD presents an intrinsic limitation, represented by the necessary modulation of immunosuppression, which is fundamental in post-transplant settings. In fact, in patients who have undergone SOT, immunosuppression is essential to prevent graft rejection, while in those receiving allo-HSCT, it plays a key role in preventing GvHD. For this reason, the reduction or withdrawal of immunosuppression in the context of CAR-T cell therapy for PTLD could be carefully managed by a multidisciplinary transplant team with specific expertise.

McKenna et al. reported the largest multicenter retrospective study to date on the use of CAR-T cells in the post-SOT setting [36]. Notably, nearly half of PTLD cases following SOT are not EBV-driven, rendering them ineligible for EBV-specific CTL therapy. This study included 22 patients with R/R PTLD post-SOT, the majority of whom (14 cases, 64%) were kidney transplant recipients. Bridging therapy was administered in 55% of patients prior to CAR-T infusion, and immunosuppression was fully discontinued in 64% of cases.

With regard to safety profile, CRS occurred in 82% of patients, with G3 and G4 events reported in one patient (5%) each. Immune effector cell-associated neurotoxicity syndrome (ICANS) was observed in 73%, including six (27%) with G3 and two (9%) with G4 neurotoxicity. The ORR was 64%, including a 55% complete response rate. Two-year PFS and OS rates were 35% and 58%, respectively. Notably, three patients (14%) experienced allograft rejection following CAR-T therapy.

Different results have been reported in limited case reports and single-center case series. Mamlouk et al. described three kidney transplant recipients, two of whom experienced relapses at 12 and 34 weeks, while one remained in remission beyond 28 weeks [37]. Similarly, Hernani et al. reported a kidney transplant recipient with PTLD who permanently discontinued immunosuppression prior to axicabtagene ciloleucel infusion [38]. At 10 months of follow-up, the patient achieved complete remission without signs of organ rejection. Luttwak et al. reported on three patients with EBV-negative PTLD, all of whom demonstrated no graft dysfunction during therapy, with responses lasting from three to nine months [39]. Conversely, Krisnamoorthy et al. documented poor outcomes in patients with kidney, heart, and pancreas transplants, all of whom developed complications from CAR-T therapy and failed to respond [40]. Finally, Portuguese et al. described a kidney transplant recipient with PTLD treated with lisocabtagene maraleucel; tacrolimus was stopped before therapy and resumed approximately two months after infusion [41]. The patient achieved CR but relapsed at eight months. No acute rejection occurred, and subsequent treatment with rituximab, ifosfamide, and radiation led to disease-free status at 11 months.

In conclusion, CAR-T cell therapy represents a promising treatment strategy for PTLD, with efficacy and safety outcomes broadly comparable to those seen in DLBCL. However, two main limitations should be considered in the use of CAR-T for R/R EBV-PTLD: the immunosuppressive state, which could nullify CAR-T efficacy, and the lymphodepleting chemotherapy administered before CAR-T infusion, which may add toxicity to the allograft patients. These complications could be carefully weighed and managed in the therapeutic decision-making process.

## 6. Discussion

EBV-positive PTLD is a rare and potentially life-threatening lymphoma that presents challenges in the management of transplanted patients. The present review aims to illustrate emerging adoptive cell immunotherapy approaches in cases of R/R EBV-driven PTLD.

Currently, the guidelines from the American Society of Transplantation (AST), European Conference on Infections in Leukemia (ECIL-6), and National Comprehensive Cancer Network Clinical Practice Guidelines in Oncology (NCCN Guidelines^®^) recognize anti-CD20 immunotherapy with rituximab both as preemptive therapy for EBV reactivation (based on EBV viral load) and for treatment of frank PTLD [4,18]. Patients who are refractory to rituximab exhibit unfavorable prognoses and do not present effective therapeutic alternatives. Although outcomes differ based on treatment protocols, rituximab-containing regimens may fail in up to 50% of individuals with EBV-positive PTLD following allo-HSCT. Several factors have been implicated in reduced responsiveness to rituximab, including the presence of acute GvHD requiring immunosuppressive therapy, extranodal disease localization, intolerance to rituximab-based immunochemotherapy, and the use of bone marrow as the stem cell source.

Treating R/R PTLD remains particularly challenging, as no standardized second-line therapy currently exists. In recent decades, adoptive immunotherapy using EBV-CTLs has emerged as a key strategy in this setting. This approach includes four main types of cell-based therapy: donor lymphocyte infusions (DLIs), autologous EBV-CTLs, and allogeneic EBV-CTLs derived either from the original donor or from third-party HLA-matched donors (“off-the-shelf” products).

DLIs have shown limited efficacy, primarily due to their high association with GvHD. The use of autologous CTLs is often hampered by the profound immunosuppression following both SOT and allo-HSCT. Donor-derived CTLs, although a feasible option in allo-HSCT, are not applicable in SOT recipients due to the allograft derived from cadaveric donors. Therefore, third-party EBV-CTLs from healthy, pre-screened HLA-matched donors—available as off-the-shelf products—have become the most practical solution.

In this context, tabelecleucel has emerged as a promising and effective treatment for patients with poor-prognosis PTLD. The phase 3 ALLELE study has confirmed its efficacy and safety profile, suggesting that, pending long-term data, this therapy could be incorporated into future treatment guidelines. A notable advantage of tabelecleucel is its ability to cross the blood–brain barrier, making it a viable option even for patients with CNS involvement.

Lastly, CAR-T cell therapy may offer a potential treatment avenue for EBV-negative but CD19-positive PTLD cases. However, its use is currently limited by a high risk of rejection, stemming from the need to reduce immunosuppression. Future improvements in balancing CAR-T activity with immunosuppressive therapy may enhance outcomes in this particularly difficult subset of patients.

## 7. Conclusions

In conclusion, EBV-specific CTLs, including the off-the-shelf product tabelecleucel, represent a promising therapeutic alternative for the treatment of relapsed/refractory EBV-driven PTLD. Accumulating clinical evidence supports their efficacy and durability of response, positioning adoptive T-cell therapy as a viable and effective option in this challenging clinical setting. Given the accumulating evidence, this therapeutic approach is expected to be integrated into future treatment algorithms and guideline recommendations for EBV-associated PTLD.

Additionally, PTLD cases that express CD30 may benefit from treatment with brentuximab vedotin. Lastly, intense clinical research is currently underway exploring the application of other targeted agents, including PD-1 inhibitors, Bruton’s tyrosine kinase inhibitors, hypomethylating agents, and inhibitors of the PI3K/Akt/mTOR signaling pathway.

## Figures and Tables

**Figure 1 antibodies-14-00047-f001:**
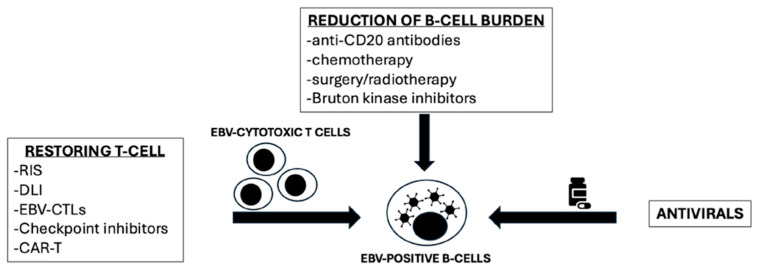
Different strategies for EBV-PTLD treatment (adapted from Styczynski et al., 2022 [5]). RIS: reduction of immunosuppression; DLI: donor lymphocyte infusion; EBV-CTLs: Epstein–Barr virus cytotoxic T-cell lymphocyte; Car-T: chimeric antigen receptor T-cell.

**Table 1 antibodies-14-00047-t001:** Summary of characteristics of different forms of EBV-CTLs. EBV-CTLs: Epstein–Barr virus cytotoxic T-cell lymphocytes; SOT: solid organ transplantation; allo-HSCT: allogeneic stem cell transplantation.

Type of EBV-CTLs	Characteristics	References
Autologous	- Low risk of graft rejection - The concomitant immunosuppressants prevent the activity of autologous EBV-CTLs - Long time in manufacturing process (3–4 months) - Difficult in production if recipient is EBV seronegative	[19,20]
HLA-matched derived from the primary donor	- Low risk of GvHD - Not possible in patients who received umbilical cord transplants or in SOT where grafts are obtained from cadavers - Required HLA-matched	[21,22,23]
Third-party products	- No limitation in SOT or allo-HSCT - Rapid access and possibility to administer multiple doses “off-the-shelf” - No influence by immunosuppressants and chemoimmunotherapy	[24,25,26,27,28]

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
