# Peer review of "Adoptive Cell Immunotherapy in Relapse/Refractory Epstein–Barr Virus-Driven Post-Transplant Lymphoproliferative Disorders"

_2073-4468, 2025, doi:10.3390/antib14020047_

Round 1

Reviewer 1 Report

Comments and Suggestions for Authors

In general, the work and study on the limited therapeutic options for importantly Post-transplant lymphoproliferative disorders (PTLD) patients with solid organ transplantation (SOT) and allogeneic hematopoietic stem cell transplantation (allo-HSCT; it is informative and readers will like to read on such poor prognostic diseases…. emerging cellular approach such as CAR T-cell therapy increases the value of such study. Thanks for the focus. Nonetheless, there are some limitations in the study and they should be more highlighted throughput the text.

-In the introduction please list the predominant organs that are largely transplant worldwide.

-For Table 1, it is strongly needed to add a column for each notions backed by reference(s); there are plenty. Please elaborate.

-L 142, …. EBV-DNA titers… please add some more information here about the titers? And clearly address how they measured and if it is protein or n.a.?

-L 146, …..experimented partially…. What does it mean?

-some abbreviations in their first appearance should be define and next part of the text should be used as abbreviated one. eg, CAR-T etc… please check and elaborate very carefully throughout the text.

-Mechanistic aspects of the effects should be better highlighted throughout the text.

-Schematically describe the mechanistic of aspects of the drugs used for SOT therapy, and show some key cells and molecules for the key findings by comparing them.

-The discussion should state the rationale for the study and the possible mechanism for the protective role of the …..
-The conclusion should be explicit, stating how the therapeutic approaches work along with future prospects, and avoid introductory words here.

-I would make a more specifically visual effective graphical abstract, to the structure of this review study and concisely main findings, with implication!

Good luck   

Author Response

-In the introduction please list the predominant organs that are largely transplant worldwide.

R: We thank the Reviewer for this valuable suggestion, which we have incorporated into the revised manuscript (highlighted in red)

-For Table 1, it is strongly needed to add a column for each notions backed by reference(s); there are plenty. Please elaborate.

R: Thank you for the suggestion. We have added a column including the relevant bibliographic references

-L 142, …. EBV-DNA titers… please add some more information here about the titers? And clearly address how they measured and if it is protein or n.a.?

R:Thank you for this clarification. In the study by Comoli et al., an EBV-DNA titer considered to indicate high risk for PTLD development was defined as 1,000 genome copies of EBV DNA per 10⁵ cells in two consecutive samples. We have accordingly added this specification to the manuscript.

As there is no universally accepted standard for nucleic acid testing assays, the ECIL-6 recommendations (Styczynski, 2016) propose threshold values for initiating preemptive therapy as 1,000 EBV copies/mL, 10,000 EBV copies/mL, or 40,000 EBV copies/mL, depending on whether EBV-DNA is measured in whole blood, plasma, or serum, respectively

-L 146, …..experimented partially…. What does it mean?

R: Thank you for the clarification. We are referring to the study by Tanzina Haque et al. (2002), which investigated the use of partly HLA-matched allogeneic EBV-specific cytotoxic T lymphocytes (EBV-CTLs) for the treatment of relapsed/refractory EBV-driven PTLD. We modified in the main text  to enhance clarity

-some abbreviations in their first appearance should be define and next part of the text should be used as abbreviated one. eg, CAR-T etc… please check and elaborate very carefully throughout the text.

R: thanks we have reported the abbreviations

-Schematically describe the mechanistic of aspects of the drugs used for SOT therapy, and show some key cells and molecules for the key findings by comparing them.

R: We thank you for your valuable suggestion. As hematologists, we had also considered dedicating more attention to the pharmacological agents used in transplantation, particularly allogeneic transplantation. However, we ultimately felt that such an in-depth discussion might go beyond the scope of the current review. That said, if you believe that addressing this aspect could enhance the manuscript, we would be pleased to include it. In that case, we kindly ask you to specify which classes of drugs you would like us to cover

-The discussion should state the rationale for the study and the possible mechanism for the protective role of the …..

R: Thank you for the clarification. We have accordingly revised the Discussion and Conclusions sections

-The conclusion should be explicit, stating how the therapeutic approaches work along with future prospects, and avoid introductory words here.

R: Thank you for the clarification. We have accordingly revised the Discussion and Conclusions sections

-I would make a more specifically visual effective graphical abstract, to the structure of this review study and concisely main findings, with implication!

R: Thank you—we fully agree with this suggestion and have prepared a graphical abstract, which we hope will meet your expectations

Reviewer 2 Report

Comments and Suggestions for Authors

The authors provide a valuable review of treatment options for EBV-PTLD, a rare complication following allogeneic hematopoietic stem cell transplantation (allo-HSCT) or solid organ transplantation (SOT), with a welcome focus on recent advancements in adoptive cell immunotherapy. While the content is largely sound and beneficial, the following suggestions are offered to enhance reader comprehension:

Line 26: Clarify "Allo-H". Please define this abbreviation for readers.

Lines 46-48 & 68-76: In lines 46-48, it's stated that for post-SOT PTLD, RIS with or without chemotherapy is the first-line treatment. However, lines 68-76 do not mention chemotherapy in this context. Please clarify this discrepancy.

Line 91: Provide a brief explanation of what CHOP chemotherapy entails.

Line 111: Please provide a brief explanation of what DHAOx entails.

Lines 209-214: The significant difference in treatment outcomes between responders and non-responders in the ALLELE study is highlighted. Please include any known information regarding the differences in baseline characteristics or predictive methods for identifying responders and non-responders.

Lines 215-240: Please clearly state the target antigen for the CAR-T cells. While it is presumed to be CD19, explicit confirmation is necessary for clarity.

Lines 259-261: Explain CAR-T graft rejection risk. Elaborate on the reasons why CAR-T cells pose a risk of graft rejection to enhance reader understanding.

Overall: Add a treatment algorithm figure. Please include a figure that illustrates the treatment algorithm proposed by the authors, which would significantly aid reader understanding of the overall treatment approach.

Author Response

Line 26: Clarify "Allo-H". Please define this abbreviation for readers.

R: Thanks for the correction, we have made the correction in the text

Lines 46-48 & 68-76: In lines 46-48, it's stated that for post-SOT PTLD, RIS with or without chemotherapy is the first-line treatment. However, lines 68-76 do not mention chemotherapy in this context. Please clarify this discrepancy.

R: Thank you for the suggestion. We have clarified this point in the main text, specifying that in post-SOT PTLD, the first-line gold standard treatment consists of a reduction in immunosuppression combined with systemic therapy. Rituximab is used in CD20-positive cases, which represent the majority, whereas chemotherapy is indicated in CD20-negative cases. If a complete or partial response is achieved, maintenance therapy with rituximab alone may be considered. In the absence of a complete response, systemic chemotherapy is strongly recommended (Jennifer E. et al., Blood, 2023).

Line 91: Provide a brief explanation of what CHOP chemotherapy entails.

R: Thanks we specified in the main text (CHOP= cyclophosphamide, doxorubicin, vincristine, and prednisone administered every 21 days)

Line 111: Please provide a brief explanation of what DHAOx entails.

R: Thanks, we specified in the main text (DHAOX= dexamethasone, high-dose cytarabine,oxaliplatin)

Lines 209-214: The significant difference in treatment outcomes between responders and non-responders in the ALLELE study is highlighted. Please include any known information regarding the differences in baseline characteristics or predictive methods for identifying responders and non-responders.

R: Thank you for the clarification. As this is a randomized clinical trial with well-defined enrollment criteria, there is homogeneity in the baseline characteristics of the patients, including age, ECOG performance status, number and type of prior therapies, as well as the morphological subtype of EBV-related lymphoma (Mahadeo et al., 2024). We have added the main baseline characteristics to the manuscript. Moreover, the authors of the study reported that no stringent predictive factors of response could be identified.

Lines 215-240: Please clearly state the target antigen for the CAR-T cells. While it is presumed to be CD19, explicit confirmation is necessary for clarity.

R: Thanks, we clarify in the main text.

Lines 259-261: Explain CAR-T graft rejection risk. Elaborate on the reasons why CAR-T cells pose a risk of graft rejection to enhance reader understanding.

R: thanks we agree with you we specified it in the text

Overall: Add a treatment algorithm figure. Please include a figure that illustrates the treatment algorithm proposed by the authors, which would significantly aid reader understanding of the overall treatment approach.

R: Thank you. We have included a therapeutic algorithm figure, which we hope will enhance the readability and overall understanding of the manuscript.